# Case study of machine learning using meteorological data

Anonymous Full Paper
Submission 36

## Abstract

Paper shows case study of machine learning carried out on meteorological data. The main goal in presented case study is to estimate rain rate over certain area using satellite data from Meteosat Second Generation (MSG). As a reference for ML process the data from network of meteorological radars located in Poland is used. Input and reference data had to be geometrically corrected and collocated before feeding it to ML process. In some variants training data was prepossessed using aggregation, which purpose was to perform data generalization. The subjects of the ML process were two empirical algorithms: Vicente and Roebeling. In presented case study also were used ML methods such as: shallow neural networks in two variants, decision trees and random forest. Experiments were conducted using data from 2015 as a training input, and data from 2016 for evaluation process. Empirical algorithms and ML models predicting rain rate in mm/h did not give satisfactory result. Training ML models to predict rain rate as order of magnitude – dBZ, gave better results than predictions in mm/h. However there is still room for improvements. Training empirical algorithms and ML methods for both mm/h and dBZ prediction on aggregated data performed much better than for unaggregated data, but tests performed after training resulted with slightly worse statistical metrics values.

## 1 Introduction

From the dawn of the applied computer science it was closely connected with meteorology. One of the firsts computers - ENIAC was used for weather forecasting [1]. Other tools which took and take significant role in weather monitoring and forecasting are satellite systems. First important meteorological satellite was TIROS-I (Television Infrared Observation Satellite I) [2, p. 15][3, Sec. I]. Application of satellites allowed to observe the state of the atmosphere from above. Also satellites introduced possibility to observe large areas in one particular moment of time. TIROS-I was a beginning of series of missions, which are continued today as JPSS satellites (Joint Polar Satellite System) in particular JPSS-2 (NOAA-21) [4, p. 314].

Later with technological advancement it became possible to build meteorological satellites designed for geostationary orbit, which is located around 5.5 Earth's radii above surface. First meteorological satellites were GEOS-1 [2, p. 16][5, p. 1-2] and Meteosat 1 [2, p. 17]. Geostationary satellites allow to monitor the weather on entire hemisphere in near real time.

A little bit earlier than satellite remote sensing the on-ground remote sensing started to rise. In 1946 first experiments with meteorological radars were carried out by Marshall et al. [6]. Today networks of meteorological radars are used to warn against dangerous storm events or monitor precipitation and winds [7].

Both geostationary systems and on-ground radars have similar data acquisition intervals, therefore data from the same moments of time could be compared. Such analysis was performed by Vicente et al. [8] and by Roebeling and Holleman [9]. Their goal was to estimate precipitation rate using data from geostationary satellite, However these studies were performed using analytical methods resulting with specialized formulas.

In this paper there will be presented a case study which incorporates previously mentioned analytical algorithms and approach with usage simple ML methods such as neural networks and decision trees. The main goal of models trained in presented experiments is to predict rain rate at specific moment using as an input data from geostationary satellite.

## 2 Materials

For the purpose of the experiment the two data sets were used: a) one from June 21th 2015 to September 23th 2015 for training process, b) second from June 1st 2016 to June 30th 2016 for evaluation purpose. Detailed composition of those data sets is described in subsection 2.1.

Mentioned earlier data sets are composed using data from two different sources. Therefore to match ML process requirements these data sets must be pre-processed first. This pre-processing is described in subsection 2.2.

In subsection 2.3 is described optional process of aggregation of input data which is used later for some variants of ML training.

### 2.1 Input Data

For studies presented in this paper two major data sources were used. The first data source are images

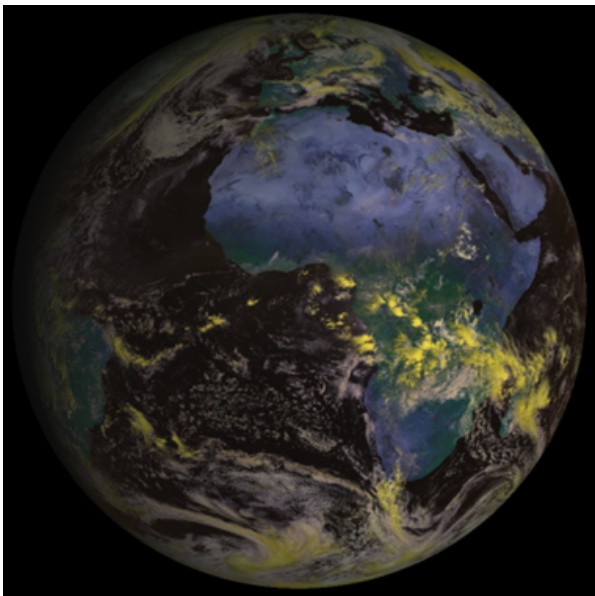

**Figure 1.** Sample image from SEVIRI sensor on Meteosat 10 located above 0° longitude. It roughly covers half of east and half of west hemisphere.

from SEVIRI sensor on board of Meteosat 10 satellite (sample RGB like image is presented on Fig. 1). SEVIRI instrument provides 11 channels spectral in resolution 3 km x 3 km in nadir and one pan-spectral channel in resolution 1 km x 1 km [10]. This raw data is available as Level 1.5 product [11]. Additionally EUMETSAT provides serveral data products computed using Level 1.5 data. One of those products is OCA (Optimal Cloud Analysis) [12]. Finally as an input ML methods following data channels were used:

- Channel 9 from Level 1.5 product - Infrared 10.8 $\mu m$,

- Cloud optical thickness from OCA or Logarithmized cloud optical thickness from OCA,

- Cloud effective particle radius from OCA.

Sample of this data is presented on Fig. 2.

The second data source is data from radars network in Poland [13, Sec. 9] provided by Institute of Meteorology and Water Management - National Research Institute [14]. Sample radar sounding as CMAX product is presented on Fig. 3. Measurements in CMAX product are expressed in $dBZ$ unit, which is logarithmic version of radar reflectivity denoted as **Z**. This relation is following:

$$Z[dBZ] = 10 \log_{10} Z \tag{1}$$

The relation between reflectivity and rain rate was described by Marshall et al. [6, Sec. 4] and is following:

$$Z = AR^\alpha \tag{2}$$

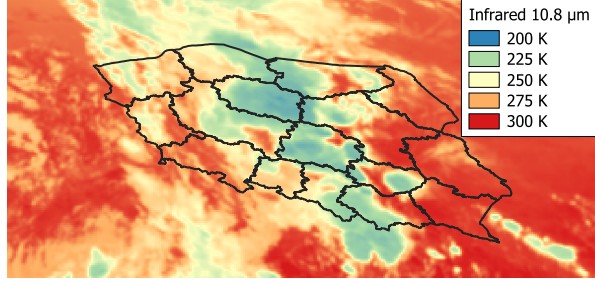

**(a)** SEVIRI Infrared 10.8 $\mu m$

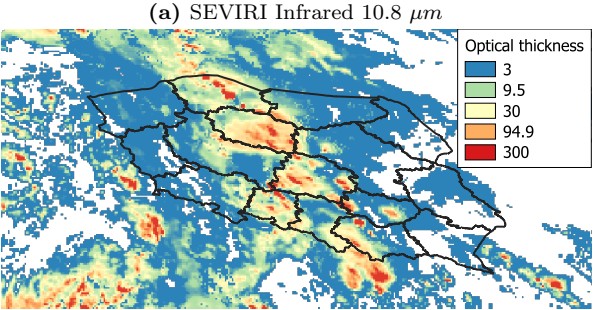

**(b)** Cloud Optical Thickness from OCA

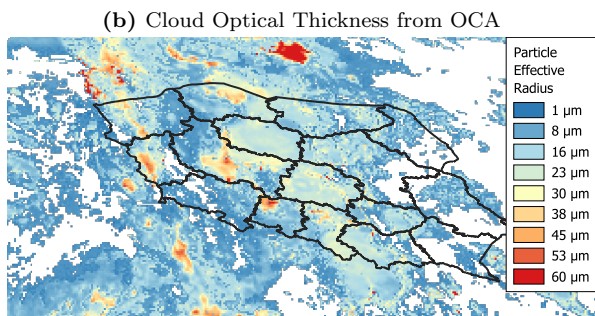

**(c)** Cloud Effective Particle Radius from OCA

**Figure 2.** Sample input data from SEVIRI Level 1.5 product and OCA product. Data covers Poland at July 25th 2015 13:00 UTC.

where **R** is a rain rate in $mm/h$, **A** is a parameter adjusted in empirical way and its suggested value for rain is 200 [15, Tab. 1], $\alpha$ is also a parameter adjusted in empirical way, and its suggested value for rain is 1.6 [15, Tab. 1].

## 2.2 Initial Data Processing

Before putting input data into ML process first some initial processing need to be done. To perform machine learning, data from satellite sensors and data from radars need to be collocated in such way that the same pixel from satellite data corresponds to the same pixel from radar data. In short terms, both selected pixels have to point to the same location. As it could be seen on Fig. 2 and Fig. 3 SEVIRI data and data from radars have different geometry. Geometry of SEVIRI images is represented by Geostationary Projection [16]. On the other hand data from radars is in Azimuthal Equidistant Projection [17]. To prevent of potential valuable data loss from radar data SEVIRI images were reprojected to Azimuthal Equidistant Projection. Reprojection

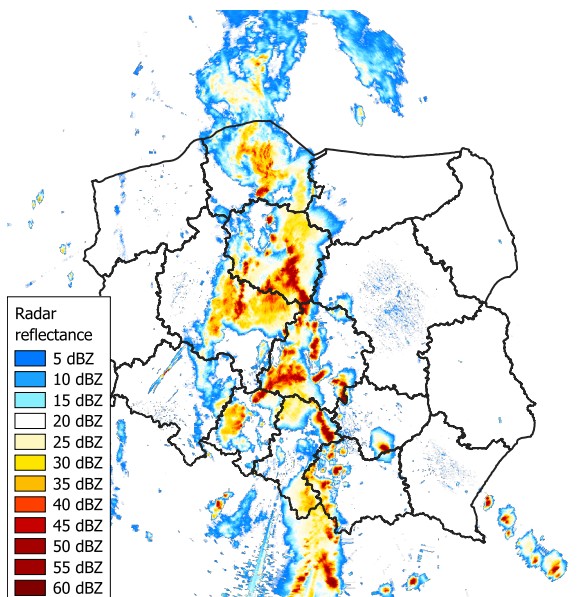

**Figure 3.** Sample radar CMAX product covering Poland at July 25th 2015 13:00 UTC.

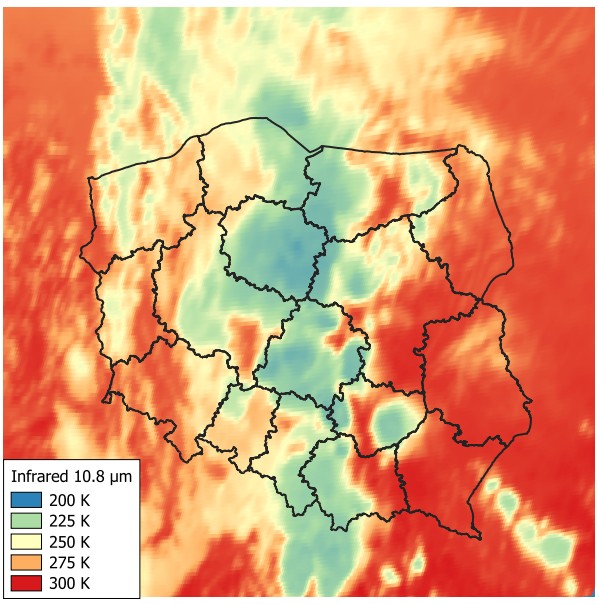

**Figure 4.** SEVIRI Infrared 10.8 $\mu m$ sample image reprojected to Azimuthal Equidistant Projection.

**Table 2.** Logarithmic quantification of input data

| Input data type | Minimal value | Upper limit | Step (Multiplier) |
| --- | --- | --- | --- |
| Cloud optical thickness | 0.0501 | < 251.1886 | 1.7783 |

## 2.3 Aggregation of training data

Because of big scattering of reference data for the same or similar values of input satellite data, for some experiments training data was a subject of aggregation process. Core idea of aggregation process is to calculate mean value of reference data for the same values of input satellite data. However because input data is in form floating point numbers, therefore searching for tuples with exactly the same input data will be pointless. Therefore to make aggregation process reasonable the input satellite data must be quantified. Quantification of each input data type is presented in Tab. 1 and Tab. 2.

## 3 Methods

For purpose of described case study two groups of models were used. First group are empirical algorithms described in subsection 3.1. Second group are ML methods briefly described in subsection 3.2.

### 3.1 Empirical Algorithms

As reference to ML methods, two state of the art solutions for rainfall rate estimation were used, those are mentioned earlier Vicente et al. [8] and Roebeling and Holleman [9]. Vicente algorithm is defined by following formula:

$$R = Ae^{BT^{\alpha}} \qquad (3)$$

**Table 1.** Quantification of input data

| Input data type | Minimal value | Upper limit | Step |
| --- | --- | --- | --- |
| Cloud top temperature | $190K$ | $< 350K$ | $1K$ |
| Cloud effective particle radius | $0\mu m$ | $< 100\mu m$ | $2.5\mu m$ |
| Logarithmized cloud optical thickness | $-1.3$ | $< 2.4$ | $0.25$ |
| Condensed Water Path (CWP) | $0g/m^2$ | $< 20000g/m^2$ | $20g/m^2$ |

was performed using `gdalwarp` program from GDAL Utilities [18]. Fig. 4 shows reprojected image from Fig. 2(a).

SEVIRI data is also burdened with error which is caused by parallax shift. Parallax shift occurs for high clouds for location far from subsatellite point [19][20]. Due this error measurements from high clouds are assigned to different location than cloud's base coordinates. In case of SEVIRI sensor this error may reach levels up to 3 pixels. Therefore to eliminate parallax shift SEVIRI data was corrected using numeric method proposed in [20, Sec. 3.2] and implemented in GEOS Height Correction program [21].

After input data is aligned temporally and spatially, that pixels from same location and same time form one data tuple. Such tuple contains all satellite input data required by algorithm or method, and contains also value from radar data as a reference for training or evaluation. Radar data depending on experiment is either in dBZ form or in rain rate in mm/h form, calculated using formula from Eq. 2.

where $\mathbf{R}$ is rainfall rate estimation in $mm/h$, $\mathbf{T}$ is top cloud temperature represented by data from Infrared 10.8 $\mu m$ channel in Kelvins and $\mathbf{A}$, $\mathbf{B}$ and $\alpha$ are adjustable parameters, their default values are defined in [8]. These parameters are marked with orange color in Eq, 3.

On the other hand there is Roebeling algorithm [9] defined by following formula:

$$R = \frac{c}{H}\left(\frac{CWP - CWP_0}{CWP_0}\right)^{\alpha} \quad (4)$$

where $\mathbf{R}$ is rainfall rate estimation in $mm/h$, $\mathbf{H}$ is height of rain column in km, $\mathbf{CWP}$ is Condensed Water Path - amount of liquid and solid water for $m^2$ of cloud in $g/m^2$. Following parameters are adjustable: $\mathbf{c}$ is scaling constant in $mm * km/h$ and its default value is $1\ mm * km/h$, $\mathbf{CWP_0}$ is a threshold for $CWP$ for which it is considered that there is a rainfall, and default value for this threshold is 180 $g/m^2$, $\alpha$ is an exponent with default value set to 0.625. Adjustable parameters are marked with orange color in Eq. 4.

$\mathbf{CWP}$ parameter from Eq. 4 is calculated using Cloud Optical Thickness and Cloud Effective Radius using formula proposed by Roebeling et al. [22, Eq. 1]. Height of rain column denoted as $\mathbf{H}$ in Eq. 4 is calculated using following formula:

$$H = \frac{CTT_{max} - CTT}{\gamma} + H_{min} \quad (5)$$

where $\mathbf{CTT}$ is Cloud Top Temperature (retrieved from SEVIRI IR 10.8 $\mu m$ channel) in Kelvins, $\mathbf{CTT_{max}}$ is maximum Cloud Top Temperature in square 128 x 128 pixels, $\gamma$ is rate of air temperature decrease for each $km$ of gained height - it is assumed to equal 6.5 $K/km$, $\mathbf{H_{min}}$ - is height of minimal rain column for thin cloud - it is assumed to equal 0.7 $km$. Equation 5 is slightly modified version of equation suggested by Roebeling and Holleman [9, Eq. 5].

All these empirical algorithms were implemented for purpose of this research using Keras ML framework [23]. Parameters marked with orange color in equations 3 and 4 are treated in Keras implementation as weights, which can be adjusted during training process.

## 3.2 Machine Learning Methods

To compare and verify performance of empirical algorithms described in subsection 3.1 some classic ML method were used in this case study to predict rain rate. First used ML method is shallow neural networks [24]. Two NN were applied:

- 25 nodes with **tanh** function in hidden layer, and 1 node with **affine** function in output layer,

**Table 3.** $R^2$ results after training of empirical algorithms and ML methods to predict rain rate in mm/h.

|  | Training | Testing | Evaluation |
|---|---|---|---|
| Vicente | 0.0354 | 0.0349 | 0.0227 |
| Roebeling | 0.0007 | 0.0007 | 0.0003 |
| NN 25 tanh, 1 affine | 0.0722 | 0.0702 | 0.0528 |
| NN 25 elu, 1 exp | 0.0724 | 0.0703 | 0.0531 |
| Decission Tree | 0.0734 | 0.0724 | 0.0447 |
| Random Forest | 0.0726 | 0.0720 | 0.0464 |

**Table 4.** $MSE$ results in $mm^2/h^2$ after training of empirical algorithms and ML methods to predict rain rate in mm/h.

|  | Training | Testing | Evaluation |
|---|---|---|---|
| Vicente | 8.57 | 8.57 | 13.78 |
| Roebeling | 10.31 | 10.59 | 16.24 |
| NN 25 tanh, 1 affine | 9.43 | 9.69 | 15.03 |
| NN 25 elu, 1 exp | 9.43 | 9.70 | 15.03 |
| Decission Tree | 9.40 | 9.66 | 15.14 |
| Random Forest | 9.41 | 9.67 | 15.11 |

- 25 nodes with **elu** function in hidden layer, and 1 node with **exp** function in output layer.

Other ML methods used in this case study are Decision Trees and Random Forest [25] in regression variant.

All ML methods are supplied with following input data:

- Channel 9 from Level 1.5 product - Infrared 10.8 $\mu m$,

- Logarithmized cloud optical thickness from OCA,

- Cloud effective particle radius from OCA.

## 4 Experiments

Two main groups of experiments were conducted for this case study. First one is predicting rain rate in mm/h, which is described in subsection 4.1. Second group is predicting order of magnitude of rain rate in dBZ described in subsection 4.2.

## 4.1 Predicting rain rate in mm/h

To predict rain rate in mm/h following empirical algorithms and ML models were trained: Vicente algorithm, Roebeling algorithm, Neural Network 25 tanh, 1 affine; Neural Network 25 elu, 1 exp; Decision Tree and Random Forest. Models and algorithms were trained on subset of training data mentioned

**Table 5.** $R^2$ results after training of empirical algorithms and ML methods to predict rain rate in mm/h. Training data was prepared using aggregation.

|  | Training | Testing | Evaluation |
|---|---|---|---|
| Vicente | 0.6306 | 0.0398 | 0.0233 |
| Roebeling | 0.0014 | 0.0031 | 0.0025 |
| NN 25 tanh, 1 affine | 0.3299 | 0.0496 | 0.0366 |
| NN 25 elu, 1 exp | 0.4651 | 0.0655 | 0.0477 |
| Decision Tree | 0.6787 | 0.0621 | 0.0394 |
| Random Forest | 0.6707 | 0.0676 | 0.0452 |

**Table 6.** $MSE$ results in $mm^2/h^2$ after training of empirical algorithms and ML methods to predict rain rate in mm/h. Training data was prepared using aggregation.

|  | Training | Testing | Evaluation |
|---|---|---|---|
| Vicente | 3.07 | 8.84 | 14.19 |
| Roebeling | 28.62 | 10.64 | 16.32 |
| NN 25 tanh, 1 affine | 3.26 | 9.94 | 15.45 |
| NN 25 elu, 1 exp | 2.68 | 9.88 | 15.41 |
| Decision Tree | 1.57 | 9.77 | 15.22 |
| Random Forest | 1.60 | 9.71 | 15.13 |

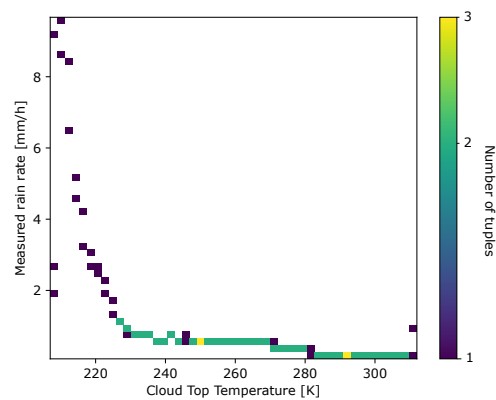

**Figure 5.** Releation between cloud top temperature and rain rate for aggregated training data used to train Vicente algorithm.

**Table 7.** $R^2$ results after training of empirical algorithms and ML methods to predict rain rate in dBZ.

|  | Training | Testing | Evaluation |
|---|---|---|---|
| NN 25 tanh, 1 affine | 0.1946 | 0.1946 | 0.2538 |
| NN 25 elu, 1 exp | 0.1937 | 0.1936 | 0.2519 |
| Decision Tree | 0.1887 | 0.1880 | 0.2372 |
| Random Forest | 0.1884 | 0.1880 | 0.2389 |

in Sec. 2, then tested on other tuples coming from the same period of time, and then evaluated on data from June 2016 also mentioned in Sec. 2.

Values of metrics $R^2$ and $MSE$ for mentioned experiments are presented in Tab. 3 and Tab. 4. Values of $R^2$ below 0.1 in Tab. 3 suggests that performance of trained models and algorithms is very poor. Especially results for Roebeling method shows that predictions are done randomly, in statistical measurements. However value of $R^2$ metric for ML models is up to two times better than Vicente which suggests that used ML methods have better ability to find correlation between input and reference data. Other interesting fact is that Vicente algorithm has the lowest values of $MSE$ metrics (Tab. 4).

To boost training process the previously mentioned experiments were rerun but using aggregated training data (see subsection 2.3). $R^2$ and $MSE$ metrics for those experiments are shown in Tab. 5 and Tab. 6. As it can be expected, values of $R^2$ for training phase increased. In case of used algorithms and ML methods in most case $R^2$ increased about 5 times. Also in most cases $MSE$ values in training phase were decreased. For the Roebeling algorithm $R^2$ increased only twice and is still on very low level. However $MSE$ for the Roebeling algorithm increased almost three times. Unfortunately in testing and evaluation phases $R^2$ and $MSE$ values for all experiments (except $R^2$ in Roebeling experiment) were slightly worse. Finally training Vicente algo-

rithm on aggregated data is equivalent to used in original publication [8, Fig. 1(a)] (see Fig. 5).

## 4.2 Predicting rain rate in dBZ

Because of unsatisfactory $R^2$ metric values for models predicting rain rate in mm/h, strategy of training was changed. Deeper data analysis showed, that relation between cloud optical thickness and radar measurements are visible when both quantities are logarithmized (see Fig. 6). Therefore to train better predictors the radar data in dBZ form was used as a reference in training process. This step may introduce apparent loose of precision in rain rate estimation, but it will help to estimate order of magnitude of precipitation phenomenon.

Empirical algorithms – Vicente and Roebeling cannot be used to estimate rain rate in dBZ, be-

**Table 8.** $MSE$ results in $dBZ^2$ after training of empirical algorithms and ML methods to predict rain rate in dBZ.

|  | Training | Testing | Evaluation |
|---|---|---|---|
| NN 25 tanh, 1 affine | 64.26 | 64.38 | 72.55 |
| NN 25 elu, 1 exp | 64.37 | 64.49 | 72.94 |
| Decision Tree | 64.74 | 64.90 | 73.80 |
| Random Forest | 64.76 | 64.90 | 73.64 |

**Table 9.** $R^2$ results after training of empirical algorithms and ML methods to predict rain rate in dBZ. Training data was prepared using aggregation.

|  | Training | Testing | Evaluation |
| --- | --- | --- | --- |
| NN 25 tanh, 1 affine | 0.5488 | 0.1455 | 0.2288 |
| NN 25 elu, 1 exp | 0.6776 | 0.1743 | 0.2318 |
| Decision Tree | 0.8441 | 0.1788 | 0.2221 |
| Random Forest | 0.8423 | 0.1808 | 0.2245 |

**Table 10.** $MSE$ results in $dBZ^2$ after training of empirical algorithms and ML methods to predict rain rate in dBZ. Training data was prepared using aggregation.

|  | Training | Testing | Evaluation |
| --- | --- | --- | --- |
| NN 25 tanh, 1 affine | 16.96 | 68.78 | 75.59 |
| NN 25 elu, 1 exp | 12.15 | 66.77 | 76.29 |
| Decision Tree | 5.83 | 65.64 | 75.27 |
| Random Forest | 5.90 | 65.47 | 75.03 |

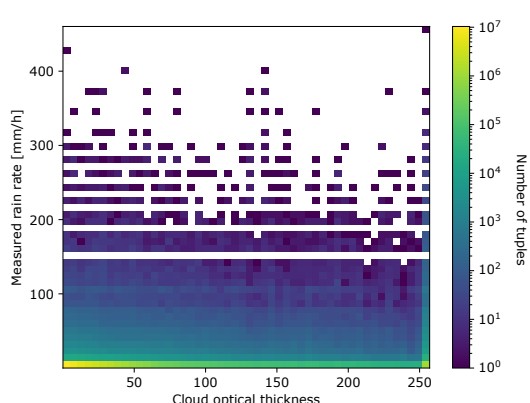

**(a)** Cloud optical thickness vs rain rate measured by radar

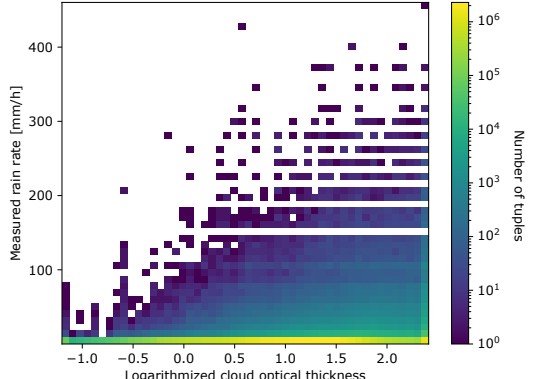

**(b)** Logarithmized cloud optical thickness vs rain rate measured by radar

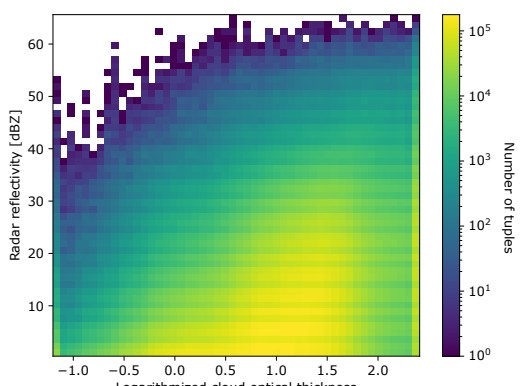

**(c)** Logarithmized cloud optical thickness vs radar reflectivity

**Figure 6.** Scatter plots of relations between cloud optical thickness and radar data in various forms.

cause these are designed to perform estimations in mm/h. Therefore dBZ estimation experiments were conducted only with ML methods. $R^2$ and $MSE$ metrics for dBZ estimation are presented in Tab. 7 and Tab. 8. $R^2$ values are about 0.1 to 0.2 higher than values for mm/h for same ML methods (see Tab. 3). Also there is surprising fact that $R^2$ for evaluation is higher than for training and testing. However $MSE$ values for same evaluation tests are higher, which is more expected behavior, because evaluation data set was not used in training phase.

Similarly to mm/h experiments, ML models were trained on aggregated input data. $R^2$ and $MSE$ values for those experiments are presented in Tab. 9 and Tab. 10. As it was expected $R^2$ values for training using aggregated data is much higher than for non-aggregated training data. For neural networks $R^2$ exceeded 0.5. $R^2$ for decision tree and random forest is very high, because values of this metric is above 0.8. Increase of efficiency of training process is also visible for $MSE$ metric values (see Tab. 10) – $MSE$ is much lower than for training without aggregation (see Tab. 8). Especially low values of $MSE$ are present for training of decision tree and random forest. Unfortunately, as before for mm/h estimation, $R^2$ and $MSE$ have slightly worse values in testing and evaluation phases for models trained on aggregated data, than for models trained on unprocessed data.

## 5 Conclusion

Presented case study shows training of empirical algorithms and ML methods for estimation rainfall rate using satellite data as input. Data from

radar network was used as reference for training process and also for testing end evaluation phases. Experiments where conducted using data prepared in various ways, such as aggregation of training data and predicting rain rate order of magnitude using radar data in logarithmic form as a reference.

This case shows that prediction of rain rate using satellite data is a difficult issue. ML training of empirical algorithms designed to estimate in mm/h did not give satisfactory results. Introducing ML methods to the case study minimally improved values of statistical metrics. Estimation of rain rate as order of magnitude in logarithmic space gave another improvement in statistical metrics. Finally the aggregation of training data made training process more effective, because some effort of data generalization was transferred to prepossessing phase. However aggregation of training data did not introduce any improvement in testing and evaluation phases. Despite that, data aggregation technique could be useful for reducing time of training phase - because of significantly lower count of data tuples.

Further improvement of this research could be made by using satellite data of higher resolution – like data from MTG satellites (Meteosat Third Generation) [26]. Other possible improvement is application of more complex ML methods such as convolutional neural network or deep learning which will help to incorporate some meteorological spatial context instead of focusing on pixel in pixel prediction.

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
