# OpenReview forum: "Case study of machine learning using meteorological data"
_NLDL.org/2026/Conference — Submitted to NLDL 2026_

### Official Review · Reviewer_YrPv · 2025-09-28
**Study about ML in rain rate forecasting that requires improvement**

**Rating:** 1
**Confidence:** 4
**Final Rating:** 1
**Final Confidence:** 5

**Summary:**

The proposed paper discusses the development of machine learning (ML) pipelines to predict rain rate in Poland. The researchers utilized satellite image data and built a data-preprocessing pipeline to align and aggregate information. Then, they proceeded to train and evaluate empirical models (Vicente and Roebeling) and four ML models: two shallow neural networks, a decision tree, and a random forest. The results show weak performance from all the models.

**Strengths:**

* The authors properly identified relevant data sources and derive pertinent features from them.
* The suggested metrics $R^2$ and MSE are suitable for the forecasting problem.
* The authors identified and implemented relevant empirical baselines to the rainfall rate estimation problem such as the so called Vicente and Holleman algorithms.

**Weaknesses:**

* The experimental design has considerable concerns, such as the weak selection of algorithms given the complexity of the problem and data. It is unclear why the authors did not compare against a CNN or vision transformers, as the data was in the form of satellite images. Moreover, the use of shallow neural networks falls short as deep learning models require further layers to extract features from images. Overall, there is no clear justification for the model selection.
* Beyond the lack of explanation on the architecture of the decision tree and random forest, it appears the ML models did not undergo any cross-validation or hyper-parameter tuning process to assess their performance and select appropriate architectures for the problem.
* The justification for the aggregation process appears naive, as it risks creating incompatible training and test sets. In other words, the model seems to be trained on different data than what it's tested on. Indeed, Tables 5 and 6 confirm this issue by showing no improvement with respect to the test set.
* Equations (1) and (2) are incompatible as Z is defined in several ways.
* Please use a more descriptive title as it does not have major relevance to the paper.
* Also, it seems there are major misconceptions about the difference between machine learning and deep learning. For instance, Convolutional Neural Networks are not only ML methods but are also a form of deep learning.
* What do the authors mean about quantifying data in section 2.3? This information is already quantitative. Do they mean quantize data?
* The paper does not present any reproducibility statement nor additional material to reproduce the experiments.
* Finally,  the paper conclusions do not align with recent literature on ML and satellite images for rainfall forecasting [1, 2], where there are empirical findings that ML models can indeed bring added value to the problem. It is strongly recommended to enhance the literature review of the project.

Reference:
[1] M. T. Brunetti, M. Melillo, S. Peruccacci, L. Ciabatta, and L. Brocca, "How far are we from the use of satellite rainfall products in landslide forecasting?," Remote Sensing of Environment, vol. 210, pp. 65–75, 2018.
[2] S.-Q. Dotse, I. Larbi, A. M. Limantol, and L. C. De Silva, “A review of the application of hybrid machine learning models to improve rainfall prediction,” Modeling Earth Systems and Environment, vol. 10, no. 1, pp. 19–44, 2023, doi: 10.1007/s40808-023-01835-x.

**Final Justification:**

As there was no rebuttal for my comment, I could not review the authors’ point of view on my comments and questions. Moreover, I acknowledge the work by the other reviewers who pointed out the lack of contributions and issues with experimental design and analysis of this work towards the AI community; thus, these points strengthen my confidence in rejecting this paper.

**Justification:**

The proposed experiment design requires a considerable reformulation, as it seems the researchers chose models with insufficient capacity to express the information from the data. Furthermore, there are missing steps in the ML development process, such as cross-validation and hyperparameter optimization. Finally, there is no incremental contribution to the field of data-driven solutions to ML for meteorology, as the paper lacks experimental rigor and a deeper discussion of related work. Overall, the paper focus too much in the technicisms rather than the show an added value from the research point of view.

---

### Official Review · Reviewer_M2z5 · 2025-10-06
**Weak language, no clear contribution**

**Rating:** 1
**Confidence:** 5
**Final Rating:** 1
**Final Confidence:** 5

**Summary:**

The paper deals with a case study in which ML applications were applied to meteorological data. The objective was to determine the Rain rate over a specific period of time. This paper shows the steps involved in data preprocessing and roughly outlines the ML models used.

**Strengths:**

The paper provides an overview of the data used and a transparent overview of the data preprocessing.

**Weaknesses:**

* Partially incomprehensible due to language deficiencies
* No clear motivation
* No state of the art
* No contribution visible
* Focus should be on ML, methodology only briefly mentioned here
* Results not examined in detail

**Final Justification:**

The authors did not provide a rebuttal, so my review stays the same:

* The paper is partially incomprehensible due to language deficiencies
* There is no clear motivation
* No state of the art
* No contribution visible
* Focus should be on ML, methodology only briefly mentioned here
* Results not examined in detail

**Justification:**

Unfortunately, the paper is very difficult to understand in many places due to language deficiencies. Furthermore, there is a complete lack of classification of the work in the current state of the art. The introduction refers to irrelevant aspects that are decades in the past instead of the current state of the art. For this reason, it is not clear what the benefit of this paper is compared to other works that already exist and which are not mentioned. A search for “case study machine learning rain fall” in Google Scholar yields numerous hits.
Furthermore, the core of the work is supposed to be a case study in the field of ML. In the section Methods, only a few lines are spent on this “focus”. The results are presented but not examined in detail or questioned.

---

### Official Review · Reviewer_ZVW7 · 2025-10-08
**Review of machine learning experiments for rain rate prediction from satellite observations**

**Rating:** 2
**Confidence:** 4
**Final Rating:** 2
**Final Confidence:** 5

**Summary:**

Paper studies the utilisation of classical machine learning regression models for rain rate prediction from Meteosat observations of SEVIRI Infrared, cloud optical thickness from OCA, and cloud effective particle radius from OCA as well as network of meteorological radars as reference data. Paper proposes pre-processing techniques, aggregation strategies, and utilisation of two physical/empirical methods from literature as well as feed forward NN, decision tree, and random forest ML methods, comparing the rain rate prediction performance in pixel-to-pixel prediction setting. Also, direct reference data and its logarithm transformed versions were evaluated. All tested methods have very low $R^2$ scores and MSE are in quite similar between the ML models as well as one of the physical models, where the latter being a simple one even beating ML models in many tests. Overall, paper has typical ML process pipeline utilising existing approaches and the modelling choices nor the results are not very convincing.

**Strengths:**

The basic idea of the paper is ok, generally problem and topic are important, and organization of the manuscript is ok. As an application and case study paper, it presents the data processing pipeline and visualisation the data, as well as baseline comparison and evaluation using the real data, showing the challenge and difficulty of the prediction problem.

**Weaknesses:**

Although data processing pipeline, methodologies, and evaluations are technically correct, there is room for improvement. Based on the results and comparison, the choice of models are not very convincing, and not showing the usefulness of ML utilisation. For example, it would have been good instead of just pixel-to-pixel regression, consider more of the structure of the observation images and possible correlating pixel information. The clarity of the paper could be improved in some parts of the pipeline description. For example, more details should be added how the aggregation processing is done. Also, the related work section is only given very generic background without analysing how proposed work related to previous approaches, especially from AI and machine learning perspectives. Overall, significance of the results are not very convincing and only standard methodologies are applied.

Questions:
- Would it be possible to transform log predicted rain rate back to the normal scale for better comparison?
- How does previously applied AI/ML techniques in this problem/topic relates to proposed case study from methodological and performance wise?

**Final Justification:**

Overall interesting machine learning problem. However, all my concerns hold after rebuttal: 1) methodological choices nor results are not very convincing, only applying very basic ML models without much novelty, and without noticeable improvements or benefits against the baseline, 2) many parts of the presentation and writing are vague and unclear, and 3) the related work analysis is very generic and there is no discussion of how proposed work compares to them and what new contributions it brings to the domain.

**Justification:**

Generally interesting ML problem, but methodological choices nor results are not very convincing, only applying very basic ML models without much novelty, and without noticeable improvements or benefits against the baseline. Some parts of the presentation are a bit vague and unclear how the data processing pipeline is currently described. Also, the related work analysis is very generic and there is no discussion of how proposed work compares to them and what new it brings to the table.

---

> ### Author Rebuttal · Authors · 2025-10-20
>
> Dear Reviewer,
> These are the answers to Your questions:
>
> > Would it be possible to transform log predicted rain rate back to the normal scale for better comparison?
>
> Those models which predicts rainfall intensity in log form, de facto predicts radar reflectivity equivalent in dBZ. To transform those predictions to `mm/h` You need to perform similar steps like in case of real radar measurements:
>
> 1. Transform reflectivity from log scale to normal scale: $Z = 10^{Z[dBZ]} / 10$ (see. eq (1))
> 2. Use relation described by Marshall et al. and calculate R: $R = (Z/A)^{1/\alpha}$ (see. eq (2)), where $A$ and $\alpha$ need to chosen experimentally or taken from literature (ie. Marshall and Pallmer 1948).
>
> >How does previously applied AI/ML techniques in this problem/topic relates to proposed case study from methodological and performance wise?
>
> For rain rate estimation using data from geostationary satellites we have studied mainly Vicente et al. 1998 and Roebeling and Holleman 2007. In those publications ML methods were not used. We are aware, that there have been recent publication, which are using more advanced neural networks like CNN for rain rate estimation.
>
> Also there are several different papers using ML and combined methods for slightly different purpose: to estimate daily or monthly accumulated rainfall. Predicting those quantities using ML perform much better than for rain rate estimation, due to accumulation over longer time span.
>
> Thank You for honest and detailed review,
> Best Regards,
> Author(s)

---

### Official Review · Reviewer_hntX · 2025-10-08
**Unclear paper studying an interesting problem**

**Rating:** 1
**Confidence:** 5

**Summary:**

This paper investigates models for rain rate estimation.

This paper explores the application of machine learning and empirical algorithms to predict rain rate using satellite data, with radar data serving as ground truth. While the problem is interesting and utilizes valuable data sources (Meteosat Second Generation and land-based radar data from Poland), and the study compares a range of methods (two empirical models: Vicente and Roebeling, along with neural networks, decision trees, and random forests), the overall contribution is significantly limited by unclear results and inconsistent methodological applications.

Data used was meteosat second generation (MSG) data and land based weather radar data from Poland. The authors have put some work into aligning the data temporally and spatially. The data is stacked into tuples, one corresponding to each location. This means that the machine learning model will have no spatial context. ML models take the satellite data as inputs and the radar data is used for ground truth.

Predicting "order of magnitude" gave better results than predicting mm/h.

Models explored: 2 different empirical models (Vicente et al. [8] and Roebeling and Holleman), 2 small neural network models, decision tree, random forest.
Data from 2015 was used as training, data from 2016 was used for evaluation.

**Strengths:**

The paper compares several methods. Both ml and engineered methods.

The problem is interesting, and the data sources could surely be modeled in interesting ways.

**Weaknesses:**

The text has grammatical errors, and does not read very easily.

The results are not very clear. The authors state that the results improve using logarithmized outputs, but the numbers are very unclear on this point. It's not clear what values can be compared between the different experiments.

It's also not very clear how the modelling was performed. For example, was the target values normalized in any way when predicting mm/h?

Table 3 shows very low R² values (below 0.1) for all methods when predicting rain rate in mm/h, indicating poor performance.
While ML models showed slightly better R² values than Vicente (up to twice), they were still very low.
For aggregated data, while R² values increased for the training phase, testing and evaluation phases still showed slightly worse R² and MSE values (Lines 289-291).
Introducing ML methods minimally improved the values of statistical metrics for rain rate estimation in mm/h.
The strategy of predicting rain rate in dBZ, while yielding better R² values, may "introduce apparent loose of precision in rain rate estimation". (Also note that the sentence is misspelled).
This implies a trade-off where the model might be better at estimating the order of magnitude but less precise in the actual rain rate value.
A "surprising fact" is noted where R² for evaluation is higher than for training and testing in dBZ prediction (Lines 315-316). While MSE values for evaluation are higher (as expected), the higher R² in evaluation could indicate overfitting on the training data, or an unusual data distribution/metric application.
SEVIRI data is subject to parallax shift, especially for high clouds, which can lead to measurements being assigned to incorrect locations.
Training data needed pre-processing using aggregation for data generalization. This adds a step and potential complexity.
Future Improvement Suggestions Imply Current Limitations:
The paper concludes by suggesting further improvements such as using satellite data of higher resolution (e.g., MTG satellites) or more complex ML methods (convolutional neural networks, deep learning). These suggestions implicitly highlight the limitations of the current approach and data used.

**Justification:**

This paper does not meet the standards required at NLDL.

A major weakness lies in the ambiguous and often contradictory presentation of results. For instance, while data aggregation notably improved R² scores during the training phase, this benefit did not translate to the testing and evaluation phases, where performance sometimes worsened. Furthermore, the R² values for predicting rain rate in mm/h are consistently very low across all methods (below 0.1), indicating the models fail to explain any meaningful variance. Even when shifting the prediction target to log space, which showed slightly better R² values, the evaluation R² was surprisingly higher than training/testing, complicating interpretation of generalization.

The paper also suffers from grammatical errors and readability issues, which hinder understanding. Crucially, the methods for processing input and output data (e.g., using log outputs) are not consistently applied or clearly articulated across different experiments, making it difficult to directly compare and evaluate the diverse methods presented.

Given the fundamental issues with the clarity and strength of the results, particularly the very low performance for the primary prediction task (mm/h rain rate) and the inconsistencies in the log space results, the paper does not demonstrate sufficient advancement or clear insights to warrant acceptance. While the problem is relevant, the current execution and presentation do not provide a convincing or robust solution.

---

### Meta-Review · Area_Chair_dZqu · 2025-11-01

**Recommendation:** Reject
**Confidence:** 4

**Metareview:**

The authors only partially submitted a rebuttal.

This paper presents a case study exploring the use of classical machine learning models and empirical algorithms to estimate rain rate from Meteosat Second Generation (MSG) satellite data, using radar observations from Poland as reference. The topic is relevant to the conference’s themes, which represent an important application area for machine learning in meteorology. However, despite this potential, the work does not meet the required standards for acceptance, according to the reviewers' feedback.

The primary reasons are a lack of depth in machine learning methodologies and insufficient validation. There is also a lack of engagement with prior literature, despite the page limit not being reached. Lastly the reviewers found the paper hard to read in general.

---

### Decision · Program_Chairs · 2025-11-05

**Decision:**

Reject

**Comment:**

Based on the reviewers and AC comments, the paper cannot be presented at the conference.